# Multiple External Root Resorption of Teeth as a New Manifestation of Systemic Sclerosis—A Cross-Sectional Study in Japan

**DOI:** 10.3390/jcm8101628

**Published:** 2019-10-04

**Authors:** Takumi Memida, Shinji Matsuda, Mikihito Kajiya, Noriyoshi Mizuno, Kazuhisa Ouhara, Tsuyoshi Fujita, Shintaro Hirata, Yusuke Yoshida, Tomohiro Sugimoto, Hiromi Nishi, Hiroyuki Kawaguchi, Eiji Sugiyama, Hidemi Kurihara

**Affiliations:** 1Department of Periodontal Medicine, Graduate School of Biomedical & Health Sciences, Hiroshima University, 1-2-3, Kasumi, Minami-ku, Hiroshima 734-8553, Japan; 2Department of Clinical Immunology and Rheumatology, Hiroshima University Hospital, Hiroshima 734-8553, Japan; 3Department of General Dentistry, Hiroshima University Hospital, Hiroshima 734-8553, Japan

**Keywords:** systemic sclerosis, multiple external root resorption, oral manifestations

## Abstract

Background: Multiple external root resorption (MERR) has been reported in systemic sclerosis (SSc) patients in Japan and Spain. To establish whether MERR is a new manifestation, we investigated the prevalence of MERR and systemic and oral manifestations to be associated with MERR in patients with SSc. Methods: Root resorption was detected by dental X-rays, panoramagraphy or cone beam computed tomography (CBCT). The prevalence of systemic and oral manifestations was examined by rheumatologists and dentists, respectively. Autoantibodies were investigated using laboratory tests. Results: MERR was detected in four out of the 41 patients (9.8%) who participated in the present study. The prevalence of digital ulcers was significantly higher in patients with MERR (MERR vs. non-MERR, 75% vs. 16.2%, *p* < 0.05), whereas that of other systemic manifestations was not. The prevalence of face skin sclerosis (100% vs. 10.8%, *p* < 0.01), calcinosis at the facial region (75% vs. 0%, *p* < 0.01), limited mouth opening (75% vs. 18.9% *p* < 0.05), temporomandibular disorder symptoms (50% vs. 2.7%, *p* < 0.05), and tongue rigidity (75% vs. 2.7%, *p* < 0.05) was significantly higher in patients with MERR. Conclusion: SSc patients with MERR had highly homogenous maxillofacial manifestations. Further clinical and basic studies are needed to elucidate the mechanisms underlying MERR in SSc patients.

## 1. Introduction

Systemic sclerosis (SSc) is a multisystem connective tissue disease that is characterized by three basic features: Immunological abnormalities, vasculopathy and fibrosis of the skin and certain internal organs [1]. It is a rare disease with a reported prevalence of 276 cases per 1,000,000 individuals in the U.S. [2]. The oral manifestations of SSc are oral mucosa atrophy, periodontal ligament (PDL) space widening, apical root resorption, dry mouth associated with Sjögren’s syndrome and calcification in the PDL space [3,4,5,6]. 

External root (cervical) resorption occurs when the body’s own immune system dissolves the tooth root structure from outside of the tooth [7]. Since methods to assess the condition of the root, including CBCT, have recently been developed, root resorption is now more easily detected. 

The causes of single root resorption include inflammation, trauma and orthodontic treatments [8,9,10]. However, the mechanisms underlying “multiple” root resorption remain unclear. The management of external root resorption is surgical debridement when invasion is still in a shallow site, such as dentine [11]. However, in the case of a deeper invasion reaching pulpal tissue, the most appropriate strategy is to monitor the tooth in follow-up visits, and when it is impossible to stop progression, tooth loss occurs. Therefore, the early detection of root resorption is necessary to protect against tooth loss. 

We and Arroyo-Bote and colleagues recently this reported multiple external root resorption (MERR) in SSc patients in Japan and Spain [12,13], respectively. These findings prompted us to examine MERR as a new manifestation of SSc. In order to elucidate disease mechanisms, all manifestations need to be investigated. Therefore, to clarify the mechanisms contributing to the pathogenesis of SSc, it is necessary to know whether MERR is a new manifestation of SSc.

In the present study, we investigated the prevalence of MERR in SSc patients. Furthermore, we evaluated systemic and oral manifestations in all subjects to identify the factors associated with MERR. 

## 2. Patients and Methods

### 2.1. Patients

This was an observational, cross-sectional study. Patients with systemic sclerosis (SSc) who routinely visit the Department of Clinical Immunology and Rheumatology in Hiroshima University Hospital, and were referred to our department between July 2015 and May 2019, were consecutively registered. Patients with an edentulous jaw were excluded. All patients fulfilled the 1980 American College of Rheumatology (ACR) classification criteria for SSc and/or the 2013 American College of Rheumatology/European League Against Rheumatism (ACR/EULAR) classification criteria for SSc [14,15].

### 2.2. Variable Assessment

The classification of limited cutaneous (lc) or diffused cutaneous (dc)-SSc and the prevalence of Raynaud’s phenomenon, microvascular disorder, skin sclerosis, ectopic calcinosis, digital ulcers, myalgia, arthralgia, gastroesophageal reflux disease (GERD), dysphagia, Sjögren’s syndrome, pulmonary fibrosis, pulmonary hypertension, cardiovascular involvements and scleroderma renal crisis (SRC), were evaluated by physician rheumatologists at the Department of Clinical Immunology and Rheumatology. Serum immunological profiles, which included anti-nuclear antibodies (ANA) and anti-centromere antibodies, anti-Scl-70 antibodies, anti-RNA polymerase III antibodies, anti-SS-A antibodies, anti-SS-B antibodies, rheumatoid factor (RF) and anti-cyclic citrullinated peptide (CCP) antibodies, were assessed at the laboratory unit in Hiroshima University Hospital.

All SSc patients were subjected to dental X-rays or panoramagraphy to find root resorption. Cone beam computed tomography (CBCT) was used when dentists needed to diagnose root resorption in three dimensions. Clinical assessments are described below.

Root resorption: Root resorption was evaluated as described previously, and a classification higher than 1Ad was defined as external root resorption [16]. More than two teeth with root resorption was defined as multiple root resorption.

PDL space widening: A widened PDL space was evaluated in dental X-ray images, as described previously [17].

Facial skin sclerosis: A smooth, tight, and expressionless face, often referred to as a “mask-like” appearance, with the simultaneous disappearance of wrinkles and perioral furrows. Nasal alae undergo atrophy, often termed a “mouse-like” face [18,19].

Temporomandibular disorder (TMD) symptoms: Patients’ symptoms, collected as categorical data (the presence or absence of TMDs), were recorded through a questionnaire investigating masticatory muscle pain at rest and chewing, neck and shoulder stiffness, temporomandibular junction arthralgia, the feeling of a locked jaw and migraines and headaches [20]. 

Reduced opening: Restricted mandibular opening of <40 mm [21].

Tongue rigidity: Lingual fibrotic induration and consequent reductions in mobility.

### 2.3. Ethical Considerations

All subjects provided informed consent for inclusion before they participated in the present study. This study was conducted in accordance with the Declaration of Helsinki, and the protocol was approved by the Ethics Committee for Epidemiological Research at Hiroshima University, Japan (approval no. E-1043; 11 December 2017). 

### 2.4. Statistical Analysis

Fisher’s exact tests were used to compare categorical variables in SSc patients with/without MERR. Odds ratios (OR) and 95% confidence intervals (95%CI) are shown below. The Student’s *t*-test was used to compare disease duration and the age of patients with or without multiple external root resorption (MERR) (MERR/non-MERR). A *p* value less than 0.05 was considered to be significant. All tests were performed using the internet-based R software package (version R 3.0.3; http://www.r-project.org).

## 3. Results

Forty-one patients (female, 85.4%) were included in the present study (Table 1). The mean age of these subjects was 62.8 ± 11.2 years (range, 42–85) with a mean disease duration of 9.6 ± 8.7 years (range, 1–40). Among all subjects, 65.9% had lc-SSc. The prevalence of systemic involvement was similar to that reported previously (musculoskeletal; 5–96% [22], gastroesophageal; 50–70% [23,24], interstitial lung; 80% [25], pulmonary hypertension; 15% [26], cardiovascular; 55% [27], and SRC; 10% [28]).

In the present study, MERR was detected in four SSc patients, including one described in a case report (pt. 1) [12]. The causes of external resorption, including trauma, periodontal and periapical inflammation, orthodontic treatment, internal bleaching or tumors, were not found in the teeth having external resorption. Dental X-rays and CBCT images showed that MERR was observed in six, six, and four teeth in pts. 2, 3, and 4, respectively (Figure 1a, Figure 2a and Figure 3a). A widening PDL space was detected in all patients. A space between canines and premolars in the upper jaw was found in pts. 3 and 4 (Figure 2a and Figure 3a). Furthermore, the deposition of calcinosis in the nasal spur was noted in pts. 1, [12], 2, and 3, while calcinosis in the palatal plate was observed in pt. 4 (Figure 2c and Figure 3c).

A summary of the results obtained on systemic or maxillofacial involvement in each patient is provided in Table 2. Two patients also had rheumatic arthritis, while another had autoimmune disease. Calcification in the PDL space was found in one MERR patient and one non-MERR patient. Furthermore, the occlusal force of pt. 1 was 77.4 N in May 2019, and was weaker than that previously reported [12].

Systemic and maxillofacial manifestations in MERR and non-MERR patients are shown in Table 3. Disease duration in MERR patients was significantly longer than that in non-MERR patients. The prevalence of digital ulcers was significantly higher in MERR patients (75%) than in non-MERR patients (16.2%, *p* < 0.05, OR = 17.0, 95%CI = 1.1–1029.3). However, the prevalence of systemic involvement was not significantly different between non-MERR and MERR patients. On the other hand, among maxillofacial manifestations, the prevalence of face skin sclerosis (100% vs. 10.8%, *p* < 0.01, OR = 127.8 [95%CI = 5.3–3106.8]), calcinosis at the facial region, (100% vs. 0%, *p* < 0.01, OR = 245.0 [95%CI = 4.1–14,556.6]), limited mouth opening (75% vs. 18.9%, *p* < 0.05 OR = 11.8, [95%CI = 0.8–693.0]), TMDs (50% vs. 2.7%, *p* < 0.05, OR = 28.3, [95%CI = 1.1–2130.3]), and tongue rigidity (75% vs. 2.7%, *p* < 0.05, OR = 75.0 [95%CI = 2.8–2023.9]) was higher in MERR than in non-MERR patients. 

## 4. Discussion

In the present study, MERR was detected in four SSc patients. These patients had higher homogeneity for several maxillofacial manifestations, but not systemic manifestations. This result suggests that there is a new group with distinctive maxillofacial manifestations in SSc.

Although multiple root resorption has been described in case reports of SSc, an epidemiological study was conducted herein for the first time, and the results obtained revealed a relationship between SSc and MERR. Only 30 cases of idiopathic MERR were reported between 1930 and 2015 [29]. Four MERR patients were identified among 41 SSc patients. This result suggests that SSc is strongly associated with the development of MERR. The occlusal force and surface area in pt. 1 was markedly weaker than that reported previously in women of the same age [30], resulting in weakened mastication ability. This may be attributed to the loss of PDL due to root resorption. To prevent this, the early detection and treatment of MERR are needed. Furthermore, pt. 3 and pt. 4 received orthodontic treatment that was not successful. A potential reason for this is the loss of PDL caused by root resorption. A careful examination of the condition of the root in SSc patients is needed prior to orthodontic treatment, and it is not recommended for MERR patients.

No significant differences were observed in the prevalence of systemic involvement between non-MERR and MERR patients, except for digital ulcers, whereas the rate of many maxillofacial manifestations was significantly higher in MERR than in non-MERR patients. Although digital ulcers are associated with systemic involvement or mortality in SSc patients [31,32,33], this is the first study to show a relationship between digital ulcers and the oral manifestation MERR. Further studies, such as a prospective study, will clarify whether digital ulcers are a strong risk factor for MERR. Differences in the prevalence of oral manifestations between SSc patients and healthy subjects have been investigated, and the prevalence of manifestations including TMDs and tongue rigidity was reported to be higher in SSc patients than in healthy subjects [18,20,34]. Furthermore, a previous study revealed a relationship between oral manifestations and systemic involvement or severity in SSc patients [35]. In the present study, MERR was found to be associated with other oral manifestations. MERR may be an important factor for assessing the severity of oral manifestations and systemic involvement. The deposition of calcinosis in the maxillofacial region was found be more frequent among MERR patients than non-MERR patients, whereas no significant differences were observed in the prevalence of subcutaneous calcinosis. Subcutaneous calcinosis in lc-SSc patients has been associated with acroosteolysis, a higher modified Rodnan skin score, and higher serum levels of phosphorus [36]. Taken together with the present results, these findings indicate that abnormalities in bone or tooth metabolism in the maxillofacial region are related to MERR. A higher prevalence of maxillofacial manifestations, such as facial skin sclerosis, mouth opening disorder, and TMDs, was found in MERR than in non-MERR patients. The root condition of SSc patients with these manifestations needs to be carefully assessed, and interdisciplinary treatment between medical doctors and dentists is needed. SSc patients with MERR may have a similar background for cell function disorders in the maxillofacial region, which implies that the mechanisms contributing to the disease may be clarified using disease-specific iPS cells of MERR patients. 

The mechanisms underlying external root resorption have been investigated, but remain unclear. Patel et al. published a review on external cervical resorption [37], and suggested three stages of resorption: Initiation, progression, and reparative. The destruction of PDL or cementum in the initiation stage triggers the formation of clastic cells, hypoxic conditions activate osteoclastogenesis in the progression stage, and finally, repaired by bone-like tissue into the resorption. 

Since MERR in SSc patients has mainly been detected in the cervical region, cementum abrasion caused by occlusal force may be a cause of MERR. In SSc patients, vasculopathy causes hypoxia in tissue [38], which indicates that hypoxic conditions in periodontal tissue, including PDL and alveolar bone, activate osteoclasts in MERR patients. Calcification in the PDL space was found in MERR and non-MERR patients. This result suggests that bone or tooth metabolism disorders lead to MERR. Furthermore, the fractured tooth in pt. 1 was histologically analyzed, and the findings obtained revealed that the fracture was repaired with bone-like tissue [12]. These findings suggest excessive bone formation in these patients is related to the resorption.

The limitations of the present study are the small sample number examined and its cross-sectional nature. A larger sample number and prospective study are needed to clarify the relationship between SSc and MERR. The root condition of SSc patients without MERR in the present study will be carefully checked for the appearance of MERR.

## 5. Conclusions

MERR was clearly identified as a manifestation in patients with SSc, and these patients had highly homogenous maxillofacial manifestations. Since MERR is a silent symptom, root conditions need to be carefully checked in patients with SSc who have maxillofacial features. Furthermore, the prevalence of SSc needs to be clarified in patients with idiopathic multiple root resorption or abnormal root conditions. Further clinical and basic studies that focus on maxillofacial involvement will contribute to the mechanisms of SSc being elucidated in more detail. 

## Figures and Tables

**Figure 1 jcm-08-01628-f001:**
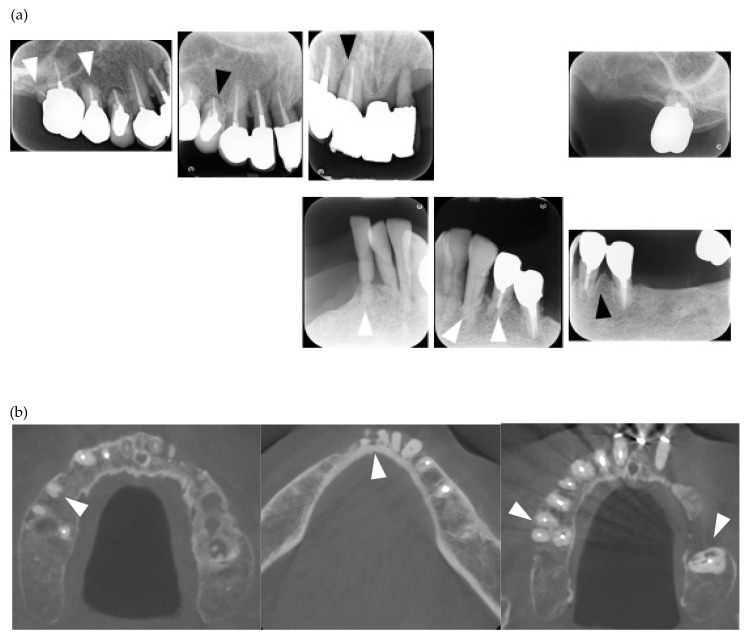
Multiple external root resorption and calcinosis at the oral region in pt. 2. A dental X-ray photo and cone beam computed tomography (CBCT) image of teeth (**a**,**b**) and the deposition of calcinosis at the nasal spur (**c**). The white arrowhead indicates root resorption, black arrowhead shows periodontal ligament (PDL) space widening and the white arrow, calcinosis.

**Figure 2 jcm-08-01628-f002:**
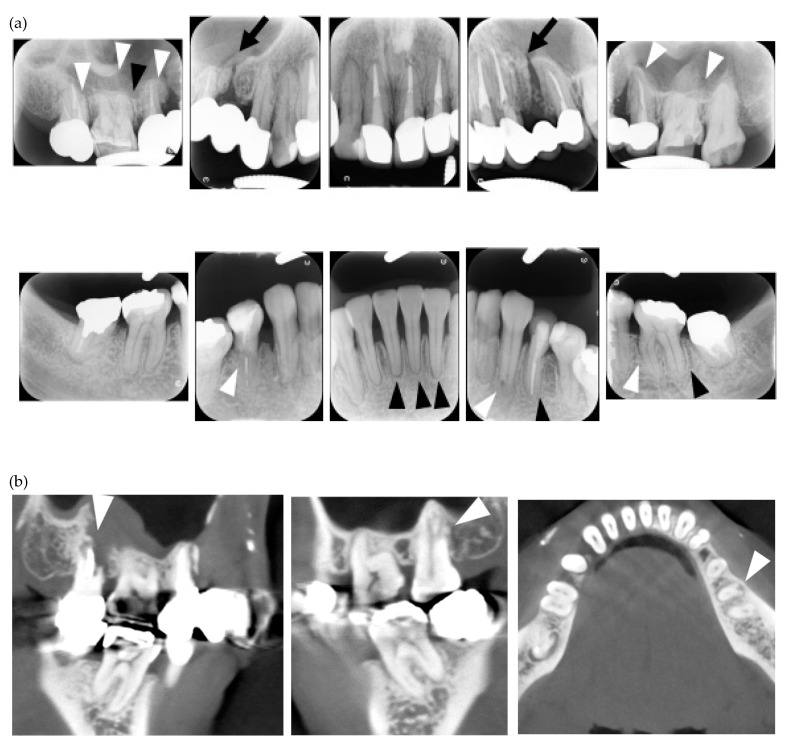
Multiple external root resorption and calcinosis at the oral region in pt. 3. A dental X-ray photo and CBCT image of teeth (**a**,**b**) and the deposition of calcinosis at the nasal spur (**c**). The white arrowhead points to root resorption, the black arrowhead to PDL space widening, the white arrow at calcinosis and the black arrow indicates the failure to close the space with orthodontic treatment.

**Figure 3 jcm-08-01628-f003:**
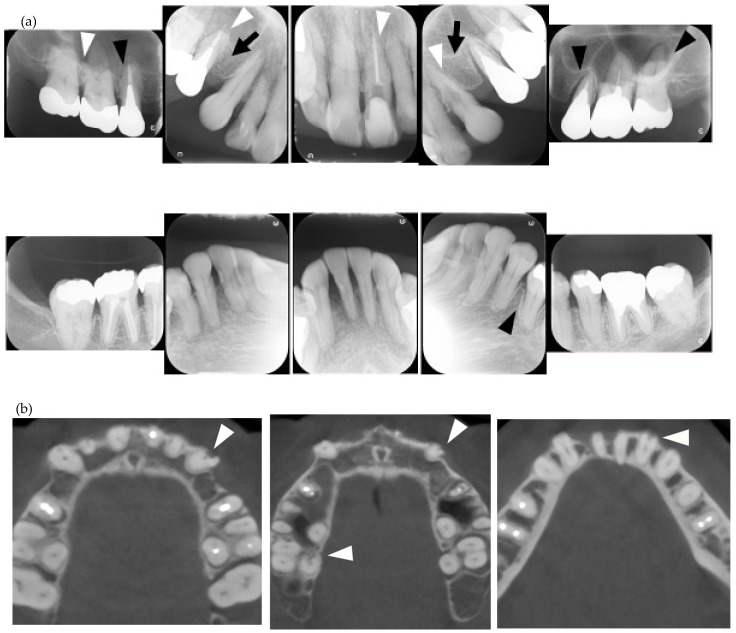
Multiple external root resorption and calcinosis at the oral region in pt. 4. A dental X-ray photo and CBCT image of teeth and the deposition of calcinosis at the palatal site (**a**–**c**). The white arrowhead represents root resorption, the black arrowhead, PDL space widening, and the white arrow, calcinosis, while the black arrow shows a failure to close the space with orthodontic treatment.

**Table 1 jcm-08-01628-t001:** Demographic data of all systemic sclerosis (SSc) patients. Systemic and maxillofacial involvement and antibodies; number (%). When the total number was less than 41, it was described as number/total number.

Demographic Data	Total (*n* = 41)
Age (mean ± SD, range, years)	62.8 ± 11.2 (42–85)
Sex (Female, %)	35 (85.4)
duration (mean ± SD, range, years)	9.6 ± 8.7 (1–40)
classification SSc (lc, %)	27 (65.9)
MERR (%)	4 (9.8)
**Microvascular disorder**	
Raynaud’s phenomenon	39 (95.1)
digital ulcers	9 (22)
**Cutaneous involvement**	
skin sclerosis	38 (92.7)
subcutaneous calcinosis	7 (17.1)
**Skeletal muscle involvement**	
arthralgia	13 (31.7)
myalgia	1 (2.4)
**Digestive involvement**	
GERD	23 (56.1)
dysphagia	9 (22)
**Respiratory involvement**	
interstitial pneumonia	25/35 (71.4)
**Cardiovascular involvement**	
cardiac insufficiency	4/27 (14.8)
pulmonary hypertension	5/36 (13.9)
**SRC**	3/19 (15.8)
**Maxillofacial manifestations**	
facial skin sclerosis	8 (19.5)
calcinosis at the facial region	3 (7.3)
limited mouse opening	10 (24.4)
Sjögren’s syndrome	23 (56.1)
TMD symptoms	3 (7.3)
PDL space widening	35 (85.4)
tongue rigidity	4 (9.8)
**Antibodies**	
anti-nuclear antibodies	37 (90.2)
anti-Scl-70 antibodies	12 (29.3)
anti-centromere antibodies	18 (48.6)
anti-RNA polymerase III antibodies	3 (7.3)

SD, standard deviation; SSc, systemic sclerosis; MERR, Multiple external root resorption; GERD, SRC, scleroderma renal crisis; TMD, Temporomandibular disorder; PDL, periodontal ligament.

**Table 2 jcm-08-01628-t002:** A summary of results on systemic and maxillofacial manifestations in all SSc patients.

					Systemic Involvement												Maxillofacial Manifestations								
Pt No.	Sex	Duration	lc/dc	MERR	RP	Dus	Skin Sc	Sub Calcin.	Arthralgia	Myalgia	GERD	Dysphagia	IP	CI	PH	SRC	Face Sc	Calcin. at Face	Limited MO	SS	TMDs	PDL SW	Tongue Rigidity	Antibody	Notes
1	F	24	lc	**Y**	**Y**	N	**Y**	**Y**	**Y**	**Y**	**Y**	**Y**	**Y**	/	/	/	**Y**	**Y**	N	**Y**	**Y**	**Y**	N	centro/RF	a case report [6]/occlusal force, 77.4N
2	F	40	dc	**Y**	**Y**	**Y**	**Y**	N	N	N	**Y**	N	**Y**	/	N	N	**Y**	**Y**	**Y**	**Y**	N	**Y**	**Y**	Scl-70/RF	
3	F	23	dc	**Y**	**Y**	**Y**	**Y**	N	**Y**	N	**Y**	**Y**	**Y**	/	/	/	**Y**	**Y**	**Y**	**Y**	**Y**	**Y**	**Y**	ANA/Scl-70/SS-A	Orthodontic treatment
4	F	11	dc	**Y**	**Y**	**Y**	**Y**	**Y**	N	N	N	N	**Y**	/	/	/	**Y**	**Y**	**Y**	N	N	**Y**	**Y**	ANA/Scl-70	calcification in PDL space/Orthodontic treatment
5	F	7	lc	N	**Y**	N	**Y**	N	N	N	**Y**	N	**Y**	N	N	/	N	N	N	**Y**	N	**Y**	N	ANA/centro	
6	M	4	lc	N	**Y**	N	**Y**	N	N	N	N	N	**Y**	N	**Y**	/	N	N	N	N	N	**Y**	N	ANA/Scl-70/RF	
7	F	16	dc	N	**Y**	N	**Y**	N	N	N	**Y**	N	**Y**	**Y**	N	/	N	N	**Y**	**Y**	**Y**	**Y**	N	ANA/RF/CCP	Rheumatic arthritis
8	F	12	lc	N	**Y**	N	**Y**	**Y**	**Y**	N	N	**Y**	N	N	N	N	N	N	N	N	N	N	N	ANA/centro	
9	F	4	dc	N	**Y**	**Y**	**Y**	N	N	N	N	N	**Y**	**Y**	**Y**	**Y**	N	N	**Y**	**Y**	N	N	N	ANA/SS-DNA/SS-A/SS-B/RF	
10	F	8	lc	N	**Y**	N	**Y**	N	N	N	**Y**	**Y**	N	N	**Y**	/	N	N	**Y**	**Y**	N	**Y**	N	ANA/RNA poly/SS-A/SS-B	
11	M	9	dc	N	**Y**	**Y**	**Y**	N	**Y**	N	**Y**	**Y**	**Y**	N	**Y**	/	N	N	N	**Y**	N	**Y**	N	ANA/centro/RF	
12	F	17	dc	N	**Y**	N	**Y**	N	**Y**	N	**Y**	**Y**	**Y**	**Y**	**Y**	N	N	N	N	**Y**	N	**Y**	N	ANA	
13	F	21	lc	N	**Y**	N	**Y**	N	N	N	**Y**	N	N	N	N	N	N	N	N	**Y**	N	**Y**	N	ANA/centro	
14	F	19	lc	N	**Y**	N	N	N	N	N	N	N	N	/	N	/	N	N	N	**Y**	N	**Y**	N	ANA/centro/AMA	
15	F	9	lc	N	**Y**	N	**Y**	**Y**	N	N	**Y**	N	N	/	N	/	N	N	N	**Y**	N	**Y**	N	ANA/centro	
16	F	1	lc	N	N	N	**Y**	N	N	N	**Y**	**Y**	**Y**	N	N	/	N	N	**Y**	**Y**	N	**Y**	N	ANA/centro	
17	F	1	lc	N	**Y**	N	**Y**	N	N	N	**Y**	N	**Y**	N	N	/	N	N	N	**Y**	N	**Y**	N	ANA	Calcification in PDL space
18	M	4	lc	N	**Y**	N	**Y**	N	N	N	N	N	**Y**	/	N	/	N	N	N	**Y**	N	N	N	ANA/Scl-70/SS-A/RF	
19	F	11	lc	N	**Y**	N	**Y**	N	**Y**	N	**Y**	N	/	/	/	/	N	N	**Y**	N	N	**Y**	N	ANA/centro/RF	
20	F	3	lc	N	**Y**	N	**Y**	N	**Y**	N	N	N	N	N	N	/	N	N	N	**Y**	N	**Y**	N	ANA/centro/RF	
21	F	4	dc	N	**Y**	N	**Y**	N	**Y**	N	**Y**	**Y**	**Y**	N	N	/	N	N	N	N	N	**Y**	N	ANA/SS-A/RF/CCP	Rheumatic arthritis
22	F	17	lc	N	**Y**	N	**Y**	N	N	N	**Y**	N	/	/	/	/	N	N	N	**Y**	N	**Y**	N	ANA/centro/RNA poly	
23	F	19	lc	N	**Y**	**Y**	**Y**	N	N	N	**Y**	N	N	N	N	N	N	N	N	**Y**	N	**Y**	N	ANA/centro/RF	
24	F	19	lc	N	**Y**	N	**Y**	N	**Y**	N	**Y**	**Y**	N	N	N	N	N	N	N	**Y**	N	**Y**	N	ANA/centro/RF	
25	F	6	lc	N	**Y**	N	**Y**	N	N	N	**Y**	N	**Y**	N	N	N	N	N	N	N	N	**Y**	N	ANA/RF	
26	F	1	lc	N	**Y**	N	**Y**	N	N	N	**Y**	N	**Y**	N	N	N	N	N	N	**Y**	N	**Y**	N	ANA/SS-A	
27	M	1	dc	N	**Y**	N	**Y**	N	N	N	**Y**	N	**Y**	N	N	N	**Y**	N	N	N	N	**Y**	N	ANA/RNA poly/CCP	
28	F	16	lc	N	**Y**	N	**Y**	**Y**	**Y**	N	N	N	N	N	N	N	N	N	N	**Y**	N	**Y**	N	ANA/centro/RF	
29	F	1	lc	N	**Y**	N	**Y**	N	N	N	N	N	N	N	N	N	N	N	N	N	N	**Y**	N	centro	
30	F	1	lc	N	N	N	N	N	N	N	N	N	N	N	N	N	N	N	N	**Y**	N	**Y**	N	centro	
31	M	20	lc	N	**Y**	N	**Y**	**Y**	N	N	N	N	**Y**	**Y**	N	N	N	N	**Y**	N	N	**Y**	N	ANA	
32	F	2	lc	N	**Y**	N	**Y**	N	N	N	**Y**	N	**Y**	N	N	N	N	N	N	N	N	**Y**	N	ANA/Scl-70/PR3	
33	F	1	lc	N	**Y**	N	N	N	N	N	N	N	N	N	N	N	N	N	N	N	N	**Y**	N	ANA/Scl-70/PR3	
34	F	2	dc	N	**Y**	N	**Y**	N	N	N	N	N	**Y**	N	N	**Y**	N	N	N	N	N	**Y**	N	ANA/Scl-70/centro	
35	M	2	dc	N	**Y**	N	**Y**	N	N	N	N	N	N	N	N	**Y**	**Y**	N	**Y**	N	N	**Y**	N	ANA/Scl-70	
36	F	6	dc	N	**Y**	**Y**	**Y**	N	**Y**	N	**Y**	N	**Y**	N	N	/	**Y**	N	N	N	N	**Y**	N	ANA/Scl-70/RF/CCP	Rheumatic arthritis
37	F	14	dc	N	**Y**	**Y**	**Y**	**Y**	N	N	N	N	**Y**	/	N	/	**Y**	N	N	**Y**	N	**Y**	N	ANA	
38	F	12	lc	N	**Y**	N	**Y**	N	**Y**	N	**Y**	N	**Y**	/	N	/	N	N	N	N	N	**Y**	N	ANA/centro	
39	F	1	lc	N	**Y**	N	**Y**	N	N	N	N	N	**Y**	/	N	/	N	N	N	N	N	**Y**	N	ANA/Scl-70	
40	F	1	dc	N	**Y**	**Y**	**Y**	N	N	N	N	N	**Y**	/	N	/	N	N	N	N	N	**Y**	N	ANA/Scl-70/SS-A	
41	F	4	lc	N	**Y**	N	**Y**	N	**Y**	N	N	N	N	/	N	/	N	N	N	N	N	N	N	ANA/centro	

Y: Yes; N: No; /: Not Checked; RP: Raynaud’s Phenomenon; DUs: Digital Ulcers; skin Sc.: Skin Sclerosis; sub calcin.: Subcutaneous Calcinosis; GERD: Gastroesophageal Reflux Disease; IP: Interstitial Pneumonia; CI: Cardiac Insufficiency; PH: Pulmonary Hypertension; face sc: Face Skin Sclerosis; Calcin. at face: Calcinosis at the Facial Region; limited MO: Limited Mouth Opening; SS: Sjögren’s Syndrome; TMDs: Temporomandibular Disorder Symptoms; PDL SW: Periodontal Ligament Space Widening; ANA: Anti-nuclear Antibodies; centro: Anti-centromere Antibodies; scl-70: Anti-Scl-70 Antibodies; RNA poly: Anti-RNA Polymerase III Antibodies; SS-A: Anti-SS-A Antibodies; SS-B: Anti-SS-B Antibodies; RF: Rheumatoid Factor; CCP: Anti-cyclic Citrullinated Peptide Antibodies; PR3: Proteinase 3 Anti-neutrophil Cytoplasmic Antibodies.

**Table 3 jcm-08-01628-t003:** Systemic and maxillofacial manifestations in multiple external root resorption (MERR) and non-MERR patients. Systemic and maxillofacial involvement and antibodies; number (%). When the total number was less than 41, it was described as number/total number. OR; Odds ratio, CI; confidence intervals.

			*p*	OR (95CI)
	MERR (*n* = 4)	Non-MERR (*n* = 37)	MERR vs. Non-MERR	MERR vs. Non-MERR
Age (mean ± SD, range, years)	50.3 ± 7.1 (43–60)	64.7 ± 10.4 (42–85)	*p* < 0.05	
Sex (Female, %)	4 (100)	31 (83.8)	N.S.	
duration (mean ± SD, range, years)	24.5 ± 10.3 (11–40)	8 ± 6.8 (1–21)	*p* < 0.01	
classification SSc (lc, %)	1 (25)	26 (70)	N.S.	
**Microvascular disorders**				
Raynaud’s phenomenon	4 (100)	35 (94.6)	N.S.	
digital ulcers	3 (75)	6 (16.2)	*p* < 0.05	17.04 (1.1–1029.3)
**Cutaneous involvement**				
skin sclerosis	4 (100)	34 (91.9)	N.S.	
subcutaneous calcinosis	2 (50)	5 (13.5)	N.S.	
**Skeletal muscle involvement**				
arthralgia	2 (50)	11 (26.8)	N.S.	
myalgia	1 (25)	0 (0)	N.S.	
**Digestive involvement**				
GERD	3 (75)	20 (48.8)	N.S.	
dysphagia	2 (50)	7 (17.1)	N.S.	
**Respiratory involvement**				
interstitial pneumonia	4 (100)	21/35 (60)	N.S.	
**Cardiovascular involvement**				
cardiac insufficiency	0 (0)	4/27 (14.8)	N.S.	
pulmonary hypertension	0/1 (0)	5/35 (14.3)	N.S.	
**SRC**	0 (0)	3/18 (16.7)	N.S.	
**Maxillofacial symptoms**				
facial skin sclerosis	4 (100)	4 (10.8)	*p* < 0.01	127.8 (5.3–3106.8)
calcinosis at the facial region	4 (100)	0 (0)	*p* < 0.01	245.0 (4.1–14,556.6)
limited mouse opening	3 (75)	7 (18.9)	*p* < 0.05	11.8 (0.8–693.0)
Sjögren’s syndrome	3 (75)	20 (54.1)	N.S.	
TMD symptoms	2 (50)	1 (2.7)	*p* < 0.05	28.3 (1.10–2130.3)
PDL space widening	4 (100)	31 (83.8)	N.S.	
tongue rigidity	3 (75)	1 (2.7)	*p* < 0.05	75.0 (2.8–2023.9)
**Antibody**				
anti-nuclear antibodies	2/2 (100)	35 (94.6)	N.S.	
anti Scl-70 antibodies	3 (75)	9 (24.3)	N.S.	
anti-centromere antibodies	1 (25)	17 (45.9)	N.S.	
anti-RNA polymerase III antibodies	0 (0)	3 (8.1)	N.S.

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
