# Peer review of "Multiple External Root Resorption of Teeth as a New Manifestation of Systemic Sclerosis—A Cross-Sectional Study in Japan"

_jcm, 2019, doi:10.3390/jcm8101628_

Round 1

Reviewer 1 Report

I was wondering why the authors performed CBCT to the patients since they have already performed periapical x-rays and panoramic Xray. There is always the ALARA or ALADA rule that we have to consider when we perform a radiographic examination. 

Moreover, I was wondering if the authors have evaluated other parameters apart from the medical history of the patients that might be related to the external root resorption. It is known that external root resorption might be caused by trauma, periodontitis, orthodontic treatment , internal bleaching, cysts, tumors, or by stimuli from a necrotic dental pulp.

Author Response

We wish to express our strong appreciation to Reviewer #1 for the insightful comments, which have helped us to improve the quality of the revised manuscript.

Reviewer comment; I was wondering why the authors performed CBCT to the patients since they have already performed periapical x-rays and panoramic Xray. There is always the ALARA or ALADA rule that we have to consider when we perform a radiographic examination. 

As reviewer pointed out, protection from radiation for patients are always necessary. And the method for CBCT in our manuscript is confusing. We did not use CBCT for all patients. Only when dentists need to evaluate in three dimensions, CBCT was used. We changed the sentence. Line 84-85.

Reviewer comment; Moreover, I was wondering if the authors have evaluated other parameters apart from the medical history of the patients that might be related to the external root resorption. It is known that external root resorption might be caused by trauma, periodontitis, orthodontic treatment , internal bleaching, cysts, tumors, or by stimuli from a necrotic dental pulp.

We deeply appreciate this constructive comment. We did not detect another causes in resorbed teeth. We added the sentence to describe this point. Line 124.

Reviewer 2 Report

This is an original paper and with great impact for clinicians, both dentists and doctors.

The final part of the introduction should be better clarified. 

minor English correction are needed.

Author Response

Reviewer comment; this is an original paper and with great impact for clinicians, both dentists and doctors.

We would like to thank the reviewer for carefully reading our manuscript and for the thoughtful comments and constructive suggestions, which have helped to improve the quality of the revised manuscript.

Reviewer comment; The final part of the introduction should be better clarified. 

As reviewer pointed out, this part is confusing for readers. We changed the sentence. Line 62-64.

Reviewer comment; minor English correction are needed.

Original manuscript was proofed by English editing service. We carefully corrected English in our manuscript again.